# Identification of QTL under Brassinosteroid-Combined Cold Treatment at Seedling Stage in Rice Using Genotyping-by-Sequencing (GBS)

**DOI:** 10.3390/plants11172324

**Published:** 2022-09-05

**Authors:** Zhifu Guo, Jialu Yao, Yishan Cheng, Wenzhong Zhang, Zhengjin Xu, Maomao Li, Jing Huang, Dianrong Ma, Minghui Zhao

**Affiliations:** 1Key Laboratory of Agricultural Biotechnology of Liaoning Province, College of Biosciences and Biotechnology, Shenyang Agricultural University, Shenyang 110866, China; 2Rice Research Institute, College of Agronomy, Shenyang Agricultural University, Shenyang 110866, China; 3Rice Research Institute, Jiangxi Academy of Agricultural Sciences, Nanchang 330200, China; 4Department of Agronomy, College of Agriculture, Purdue University, West Lafayette, IN 47907, USA

**Keywords:** rice, brassinosteroids, cold tolerance, QTL mapping, candidate genes

## Abstract

Cold stress is a major threat to the sustainability of rice yield. Brassinosteroids (BR) application can enhance cold tolerance in rice. However, the regulatory mechanism related to cold tolerance and the BR signaling pathway in rice has not been clarified. In the current study, the seedling shoot length (SSL), seedling root length (SRL), seedling dry weight (SDW), and seedling wet weight (SWW) were used as the indices for identifying cold tolerance under cold stress and BR-combined cold treatment in a backcross recombinant inbred lines (BRIL) population. According to the phenotypic characterization for cold tolerance and a high-resolution SNP genetic map obtained from the GBS technique, a total of 114 QTLs were identified, of which 27 QTLs were detected under cold stress and 87 QTLs under BR-combined cold treatment. Among them, the intervals of many QTLs were coincident under different treatments, as well as different traits. A total of 13 candidate genes associated with cold tolerance or BR pathway, such as BRASSINAZOLE RESISTANT1 (OsBZR1), OsWRKY77, AP2 domain-containing protein, zinc finger proteins, basic helix-loop-helix (bHLH) protein, and auxin-induced protein, were predicted. Among these, the expression levels of 10 candidate genes were identified under different treatments in the parents and representative BRIL individuals. These results were helpful in understanding the regulation relationship between cold tolerance and BR pathway in rice.

## 1. Introduction

Rice (*Oryza sativa* L.) is one of the most important food crops that grows in tropical to temperate regions worldwide [1]. Low temperature is a severe environmental restriction that most strongly impacts rice growth and development, especially at the early seedling and reproductive stages [2]. Cold stress at the early vegetative stage can result in stunted growth and increased seedling mortality, which in turn, leads to uneven seedling stand establishment, delayed panicle development, spikelet sterility, and subsequently, decreased yield of rice [3,4]. Therefore, it is a crucial object in rice cultivation and breeding to improve cold tolerance at seedling stage.

Cold tolerance of rice is a complex quantitative trait controlled by multiple quantitative trait loci (QTL), which is directly related to a large amount of physiological and biochemical processes and environmental factors [3,4]. To date, numerous backcross inbred lines, recombinant inbred lines, and near-isogenic lines have been developed using many cold tolerance varieties as donors, and a massive number of cold tolerance-related QTLs have been identified on 12 chromosomes at different development stages in rice [5,6]. However, these are not sufficient to mine the useful genes and reveal the molecular mechanisms of cold tolerance. In recent years, the methods to obtain a single nucleotide polymorphism (SNP) molecular marker based on the high-throughput sequencing technologies, such as genotyping-by-sequencing (GBS), genome-wide association studies (GWAS), and bulked segregant RNA-seq (BSR), are increasingly used to identify QTLs across the whole genome [7,8]. In particular, as GBS technology is rapid, simple, and easy to implement for QTL mapping, genomic selection, and genetic diversity, it has been successfully applied in QTL mapping for cold tolerance in rice [9,10].

Dongxiang wild rice (*Oryza rufpogon* Grif, DXWR) is a relative ancestor of the cultivated rice, which can be used as a donor of novel and favorable alleles for rice breeding [11,12,13]. DXWR has a high tolerance to cold stress at all growth stages, and its underground stem can tolerate temperatures up to −12.8 °C [14]. Many QTLs associated with cold tolerance have been identified in DXWR [15,16,17,18,19]. Therefore, DXWR is an elite material for enhancing the cold tolerance of rice by hybridization, backcrossing or genetic transformation.

Brassinosteroids (BR) are important steroid hormones, which play vital roles in the growth, development, and tolerance to various stresses in plants. The exogenous BR application can enhance low-temperature tolerance in plants [20,21,22]. It was shown that BRASSINOSTEROID INSENSITIVE 1 (BRI1) and BRASSINAZOLE RESISTANT1 (BZR1) positively regulate cold tolerance, whereas BRASSINOSTEROID INSENSITIVE 2 (BIN2) negatively regulates cold tolerance in *Arabidopsis*
*thaliana*. In addition, several genes related to BR biosynthesis, *BR6ox2*, *DWF4*, and *CPD*, are rapidly downregulated under low-temperature stress [23,24,25,26]. In rice, the treatments with BR can also effectively improve cold tolerance at germination and seedling stages [27,28,29,30]. To date, however, few genes associated with both cold tolerance and BR treatments in rice have been studied. The regulatory relationship between cold tolerance and BR remain largely unclear.

In the current study, a backcross recombinant inbred lines (BRIL) population of 140 individuals, derived from a cross between the DXWR and a super rice SN265, was constructed. Based on this BRIL, a high-resolution genome-wide SNP genetic map was developed using the GBS technique. Furthermore, we identified some QTLs and candidate genes under cold stress and BR-combined cold treatment at the seedling stage. The results have an important significance for the discovery of the genes involved in cold tolerance and for understanding the molecular mechanism that regulates the cold tolerance related to the BR signaling pathway in rice.

## 2. Results

### 2.1. Phenotypic Characterization under Cold Stress and BR-Combined Cold Treatment

Cold tolerance of each BRIL individual was evaluated under cold stress and BR-combined cold treatment at the seedling stage. The BRIL population showed varying levels of cold tolerance and significant contrasting response in SSL, SRL, SDW, and SWW (Figure 1). SSL was in the range of 20–40 cm and the majority of BRIL individuals were distributed in the range of 30–35 cm under normal temperature. Meanwhile, SRL was in the range of 3–9 cm and the majority of BRIL individuals were concentrated in the range of 4–7 cm. After cold stress and BR-combined cold treatment, the SSL of the majority of BRIL individuals was significantly reduced. SSL was in the range of 10–35 cm and the majority of BRIL individuals were distributed in the range of 20–30 cm. Compared to cold stress, the number of BRIL individuals with SSL in the range of 25–30 cm significantly increased under the BR-combined cold treatment (Figure 1b). After cold stress and BR-combined cold treatment, the SRL of the majority of BRIL individuals also reduced; however, it was less pronounced than the SSL. The number of BRIL individuals with SRL in the range of 4–5 cm, 6–7 cm, and 8–9 cm decreased, respectively. In addition, the SRL difference between cold stress and BR-combined cold treatment was not significant (Figure 1c).

For SDW and SWW, both cold stress and BR-combined cold treatment resulted in a significant decrease in SDW and SWW of the majority of BRIL individuals. The change in the number of BRILs with SDW in the range of 0.01–0.03 g and SWW in the range of 0.1–0.2 g were remarkable, respectively. Compared to cold stress, the number of BRILs with SDW in the range of 0.02–0.03 g and SWW in the range of 0.15–0.2 g increased under the BR-combined cold treatment, respectively (Figure 1d,e).

The correlation among all four traits with different treatments in the BRIL population was analyzed. The results revealed that all traits showed continuous and approximately normal distributions. Most of the traits showed a significant positive correlation regardless of cold stress or BR-combined cold treatment (Figure 2). For example, the significant positive correlations were observed between SSL and SRL (r = 1.00, *p* ≤ 0.01), SSL and SDW (r = 0.29, *p* ≤ 0.01), SSL and SWW (r = 0.31, *p* ≤ 0.01), SRL and SDW (r = 0.29, *p* ≤ 0.01), SRL and SWW (r = 0.31, *p* ≤ 0.01), and SDW and SWW (r = 0.85, *p* ≤ 0.01) under cold stress, respectively. Under BR-combined cold treatment, the positive correlations between SSL and SRL (r = 1.00, *p* ≤ 0.01) as well as between SDW and SWW (r = 0.57, *p* ≤ 0.01) were shown similar to under cold stress. Additionally, under both cold stress and BR-combined cold treatment, all of the SSL, SRL, SDW, and SWW showed highly significant positive correlations with r = 0.19 (*p* ≤ 0.1), r = 0.19 (*p* ≤ 0.1), r = 0.43 (*p* ≤ 0.01), and r = 0.76 (*p* ≤ 0.01), respectively. These results suggested that all four traits showed continuous single-peak pattern distributions, which met the requirements for QTL mapping. Moreover, the treatment of BR improved the SSL, SDW, and SWW of some BRILs under cold stress.

### 2.2. Construction of the Linkage Maps

Through the GBS approach, a total of 64.48 Gb high-quality filtered reads were filtered and 96.19% of these reads were mapped to the rice reference genome. Furthermore, a total of 10,836 SNPs were validated for the determination of recombinant events. The ratio of Q20 for BRILs was above 90% and the guanine-cytosine (GC) content was 43.55%, thus the quality of the data met the requirements for further analysis. A total of 10,836 unfiltered SNPs were validated for the determination of recombinant events. Finally, after filtration, followed by the ABH plugin, a total of 1145 bin markers were retained and used in the linkage map construction using the polymorphic markers to map all 12 rice chromosomes. All 12 linkage groups varied in the number and density of markers, with a total of 3188.33 cM in genetic distance and the individual linkage groups ranged from 186.74 to 346.46 cM in length. The interval of genetic distance of each polymorphic marker ranged from 1.9 to 146.3 cM and the average genetic distance between markers was 0.3 cM. The highest number of markers was 117 on chromosome 4, while the lowest number was 77 on chromosome 7 (Table 1).

### 2.3. Identification of QTLs for Cold Tolerance

The genetic linkage map was used to map the QTLs related to cold tolerance based on SSL, SRL, SDW, and SWW, respectively. A total of 114 QTLs were identified, of which 27 QTLs were detected under cold stress and 87 QTLs under BR-combined cold treatment. The maximum LOD score was 11.4737, while the maximum phenotypic variation explained (PVE) value was 18.2439% (Figure 3).

For SSL, a total of 9 QTLs with LOD scores ranging from 3.1103 to 5.0857 and PVE values ranging from 4.1682% to 10.7271% were identified on chromosomes 1, 2, 3, 4, 7, and 11, respectively. Among them, 3 QTLs (*qSSL1-2*, *qSSL3-1*, and *qSSL11-1*) were localized in cold stress. Under BR-combined cold treatment, 4 QTLs (*qSSL1-1*, *qSSL2-1*, *qSSL4-1*, and *qSSL11-2*) were identified using SSL under normal temperature condition as control, while 2 QTLs (*qSSL2-2* and *qSSL7-1*) were localized using SSL under cold stress as control (Table 2 and Appendix A).

For SRL, only 2 QTLs (*qSRL10-2* and *qSRL11-2*) were identified on chromosome 10 and 11 under cold stress. Under BR-combined cold treatment, only 2 QTLs (*qSRL1-3* and *qSRL8-1*) were localized on chromosomes 1 and 8 using SRL under normal temperature condition as control, while 29 QTLs were localized on all chromosomes except for chromosome 8 with the control of SRL under cold stress. The LOD scores of all QTLs ranged from 3.0047 to 4.3086 and the PVE values ranged from 0.7550% to 9.4108% (Table 3 and Appendix A).

For SDW, a total of 49 QTLs with LOD scores ranging from 3.0094 to 11.4737 and PVE values ranging from 0.4007% to 6.5896% were localized on all chromosomes except for chromosome 7. Among these, 19 QTLs were localized under cold stress. Under BR-combined cold treatment, 21 QTLs were localized using SDW under normal temperature condition as control, while 9 QTLs were localized using SDW under normal temperature condition as control. Interestingly, the intervals of many QTLs under different treatments were coincident. For example, there were 12 pairs of QTLs between cold stress and BR-combined cold treatment with the control of SDW under normal temperature condition, such as *qSDW1-3* and *qSDW1-4*, *qSDW6-1* and *qSDW6-2*, *qSDW6-11* and *qSDW6-12* etc. Similarly, there were two pairs of QTLs between BR-combined cold treatment with the control of SDW under normal temperature condition and those using SDW under cold stress as control, including *qSDW2-1* and *qSDW2-2*, as well as *qSDW6-9* and *qSDW6-10*, respectively. In addition, even intervals of 3 QTLs under three different treatments were coincident, including *qSDW6-1*, *qSDW6-2*, and *qSDW6-3*, as well as *qSDW6-4*, *qSDW6-5*, and *qSDW6-6*, respectively (Table 4 and Appendix A).

For SWW, a total of 23 QTLs with LOD scores ranging from 3.1392 to 8.4626 and PVE values ranging from 0.7134% to 18.2439% were identified. Among them, only 3 QTLs (*qSWW1-3*, *qSWW7-1*, and *qSWW12-2*) were localized under cold stress. Under BR-combined cold treatment, 5 QTLs were localized with the control of SWW under normal temperature condition, while 15 QTLs were localized with the control of SWW under cold stress. Similar to SDW, the intervals between several pairs of QTLs under cold stress and BR-combined cold treatment were also coincident, including *qSWW6-1* and *qSWW6-2*, *qSWW6-3* and *qSWW6-4*, and *qSWW6-6* and *qSWW6-7*, respectively (Table 5 and Appendix A).

Interestingly, intervals between some QTLs related to different traits were also coincident, including *qSRL6-2* and *qSDW6-4/qSDW6-6*, *qSRL6-1* and *qSDW6-2*, *qSWW1-1/qSWW1-2* and *qSDW1-1/qSDW1-3*, *qSWW6-2* and *qSDW6-5/qSDW6-6*, *qSWW6-3* and *qSDW6-7*, *qSWW6-6* and *qSDW6-13*, *qSWW5-1* and *qSDW5-1*, *qSWW12-4* and *SDW12-5*, *qSWW9-1* and *qSDW9-5*, *qSWW10-1* and *SDW10-1*, and *qSDW11-1* and *qSWW11-1*, respectively.

### 2.4. Prediction of Candidate Genes

The QTL regions with coincident intervals and higher LOD (more than 4.0) were used to predict candidate genes based on the Rice Genome Annotation Project website. A total of 156 candidate genes were predicted, of which 65 were annotated with known function, while the other genes were identified as retrotransposon and transposon proteins (40), expressed proteins (44), and hypothetical proteins (7), respectively (Appendix A).

Among the 65 annotated genes, 13 were possibly involved in cold resistance or BR signal transduction pathway (Table 6). They included several transcription factors, including OsWRKY77 (LOC_Os01g40260), AP2 domain-containing protein (OC_Os06g44750), MYB (LOC_Os11g45740 and LOC_Os05g51160), zinc finger proteins (C3HC4 type, LOC_Os05g39380; C2H2 type, LOC_Os09g27650; LSD1 type, LOC_Os12g41700), basic helix-loop-helix (bHLH) transcription factor (LOC_Os07g08440), and BRASSINAZOLE RESISTANT1 (OsBZR1, LOC_Os07g05805), and some other proteins, such as auxin-induced protein (LOC_Os06g01966), oxygen evolving enhancer protein (LOC_Os07g01480), and heat shock protein (LOC_Os12g41820). It should be mentioned that BZR1 is a critically important transcription factor in BR signal transduction pathway as well as cold response pathway in plants.

### 2.5. Validation of Candidate Genes

To further validate functions of the candidate genes, qRT-PCR was used to determine the expression levels of the 10 candidate genes under cold stress and BR-combined cold treatment in the parents and representative BRIL individuals (three excellent cold tolerance and three non-excellent cold tolerance).

The expression levels of all transcription factor genes, including LOC_Os01g40260 (OsWRKY77), LOC_Os11g45740 (MYB), LOC_Os07g05805 (OsBZR1), LOC_Os07g08440 (bHLH), LOC_Os06g44750 (AP2), and LOC_Os05g39380 (zinc finger), were higher under both cold stress and BR-combined cold treatment than those under normal temperature in all of the materials mentioned above. For different treatments, the expression levels of LOC_Os07g05805 (OsBZR1) and LOC_Os07g08440 (bHLH) under BR-combined cold treatment were higher than those under cold stress. For the maternal and paternal parents, the expression levels of LOC_Os11g45740 (MYB) and LOC_Os07g08440 (bHLH) were higher in DXWR than in SN265 under both cold stress and BR-combined cold treatment, whereas the opposite expression pattern was monitored for LOC_Os07g05805 (OsBZR1). For the representative BRIL individuals, the expression levels of LOC_Os11g45740 (MYB), LOC_Os07g05805 (OsBZR1), and LOC_Os07g08440 (bHLH) in three excellent BRIL individuals were higher than those in three non-excellent BRIL individuals under cold stress as well as under BR-combined cold treatment, whereas the opposite expression pattern was detected for LOC_Os01g40260 (OsWRKY77) (Figure 4). In addition to the candidate genes mentioned above, the expression levels of other genes LOC_Os06g01966 (auxin-induced protein), LOC_Os12g41820 (heat shock protein), LOC_Os07g01480 (oxygen evolving enhancer protein), and LOC_Os09g27660 (F-box protein) were not affected by cold stress and BR-combined cold treatment (Appendix A).

## 3. Discussion

The rich genetic diversity is an important basis for improving the agronomic trait and stresses tolerance in plants. Previous studies regarding evolution and genetic divergence showed that cultivated rice was domesticated from wild rice species. However, during the domestication process with thousands of years of evolution and natural selection, the diversity of many morphological traits was reduced, and many gene resources were lost in cultivated rice [14,31,32,33]. As the relative ancestor of the cultivated rice, common wild rice has high diversity of desirable genes, which are very important genetic resources for improving the agronomic trait and stresses tolerance in cultivated rice [34,35,36]. DXWR (*O. ru**fi**pogon*) was discovered in Dongxiang County, Jiangxi Province, China, which has the northernmost distribution in all species of wild rice (28°14′ N). Since DWXR plays important roles in basic rice research and industrial development, it is known as “the panda of wild plants” [17,18,37]. DXWR possesses abundant genetic resources associated with wide cross-compatibility, fertility restoration, cytoplasmic male sterility, high grain yield, and resistance to a range of biotic and abiotic stresses, especially low-temperature tolerance.

Although wild rice and cultivated rice have the same genome type, they exhibit many differences in their genome sequences. If the temporary genetic populations, such as F_2_ or BC_1_ populations and the permanent primary genetic populations, such as recombinant inbred line were constructed and used for QTL mapping, they could show not only a low QTL detection efficiency, but also poor stability. Therefore, a stable and reliable mapping population is essential for utilizing wild rice [32,33,38]. To date, multiple types of mapping populations have been constructed using DXWR as paternal parent, while a large number of QTLs associated with cold tolerance at the germination, seedling, booting, and flowering stages have been mapped to all chromosomes [16,17,18,19,30]. In this study, the female parent SN265 was used to backcross for an additional four times after a cross between the female parent SN265 and female parent DXWR (BC_4_F_1_). Furthermore, the BC_4_F_1_ population was self-crossed for eight times to obtain advanced BRIL populations (BC_4_F_8_). The back- and self-crossing for multiple generations reduced the genetic differences caused by different genetic bases between cultivated rice and wild rice, which made the result of QTL mapping more stable and reliable.

It is well known that the low-temperature tolerance of plants is a complicated quantitative trait controlled by polygenes. To cope with low-temperature stress, plants modify their physiology, metabolism, and growth by means of signal transduction and expression regulation of many genes associated with cold tolerance when plants are exposed to low-temperature stress [5]. A substantial number of genes that facilitate cold signaling and control the expression of cold regulons have been identified in many plants. A cascade signaling pathway, ICE–CBF–COR, is the most intensively studied, and is thought to be pretty important for cold tolerance in plants. This pathway contains the core components CBF/*DREB* (C-repeat binding factors/dehydration-responsive element-binding proteins) transcriptional factor, ICE (inducer of CBF expression) activated factor, and diverse downstream functional proteins called cold-regulated (COR) proteins. The CBFs belong to the AP2/ERF (apetala 2/ethylene response factor) family of transcription factors, which are the most important in the ICE–CBF–COR pathway. The CBFs control the expression of *COR* genes in response to cold stress, while the ICE acts as a positive upstream regulator of *CBF* genes [39,40,41].

In rice, 10 *CBF/DREB* homologous genes (*OsDREB1A-OsDREB1J*) have been identified. Among them, *OsDREB1A*, *OsDREB1B*, *OsDREB1D*, *OsDREB1F*, and *OsDREB1G* enhanced cold tolerance in Arabidopsis or rice [42,43]. In the case of QTL mapping studies, Os*DREB1G* and *OsDREB1J* were identified as the candidate genes related to cold tolerance for three QTLs, *qLOP2*, *qPSR2*-*1* and *qSR8-3*, using the DXWR populations as the donor parent, respectively [16,30]. In addition, an ICE1-like gene, *OrbHLH001*, was isolated from DXWR, which enhanced cold tolerance when expressed in transgenic Arabidopsis [44]. Apart from these core components, many other activators or repressors are related to this pathway that directly or indirectly affect the low-temperature tolerance of plants. A MYB transcriptional factor, MYB15, interacted with ICE1 and was bound to MYB recognition sequences in the promoters of CBF genes, which repressed the expression of CBF genes. ICE1 activated the transcription of CBF3 by binding to MYC recognition elements in the promoter [39,45]. *OsbHLH002*, a homolog of ICE1 in rice, positively regulated cold tolerance by promoting the expression of OsTPP1, which encoded a key enzyme for trehalose biosynthesis [46]. Two F-box proteins, EIN3-binding F-box 1/2 (EBF1/2), positively regulated the expression of *CBF* genes by regulating the degradation of EIN3 and PIF3 [47,48]. In the current study, we identified some genes similar to those described above, such as bHLH transcription factor (LOC_Os07g08440), F-box and another domain-containing protein (LOC_Os09g27660 and LOC_Os10g04590), MYB family transcription factor (LOC_Os11g45740), and AP2 domain-containing protein (LOC_Os06g44750). The transcript levels of these genes except for LOC_Os09g27660 and LOC_Os10g04590 (F-box) were upregulated both under cold stress and BR-combined cold treatment. In the representative BRILs, LOC_Os07g08440 (bHLH), LOC_Os11g45740 (MYB), and LOC_Os06g44750 (AP2) showed higher transcript levels in the excellent BRILs than the non-excellent BRILs under cold stress and BR-combined cold treatment.

Previous studies have shown that the exogenous application of BR can improve low-temperature tolerance at different growth stages in various plants, including rice [27,28], maize [49], wheat [50], winter rye [51], and Arabidopsis [24,26], etc. To date, the molecular mechanisms regarding both cold stress and BR signaling pathway were only intensively studied in Arabidopsis. Moreover, the cold tolerance in BR signaling pathway was involved in ICE–CBF–COR signaling pathway. Under cold stress, BR directed a bHLH transcription factor CESTA (CES) to regulate the expression of the CBF and downstream COR genes in Arabidopsis [45]. BZR1 was a very important transcription factor in BR signaling pathway, acting upstream of CBF1 and CBF2 to directly enhance low-temperature tolerance in Arabidopsis. Interestingly, OsBZR1 (LOC_Os07g05805) was predicted for *qSSL7-1* under BR-combined cold treatment in the current study. The expression level of OsBZR1 was upregulated both under cold stress and BR-combined cold treatment, and it showed higher transcript levels in the excellent BRILs than the non-excellent BRILs under cold stress and BR-combined cold treatment. Moreover, BZR1 regulated other genes uncoupled with CBFs, such as WKRY6, PYL6, and SOC1, etc., to improve cold tolerance in Arabidopsis [24,26]. As a negative modulator in BR signaling pathway, the protein kinase BIN2 interacted with and phosphorylated ICE1 under cold stress, which facilitated the interaction between ICE1 and the E3 ubiquitin ligase HIGH EXPRESSION OF OSMOTICALLY RESPONSIVE GENE1 (HOS1), and thereby promoted ICE1 degradation. It was suggested that BIN2 mainly downregulated ICE1 abundance when the expression levels of *CBF* genes were attenuated [26]. In this study, OsWRKY77 (LOC_Os01g40260), a WRKY transcription factor, was also identified for qSSL1-2 under BR-combined cold treatment. It was shown that the expression level of *OsWRKY77* was upregulated in the sensitive genotype, suggesting its negative role in cold tolerance at the germination stage in rice [52]. The expression levels of *OsWRKY77* were upregulated both under cold stress and BR-combined cold treatment, and its expression levels in the excellent BRILs were lower than the non-excellent BRILs under cold stress and BR-combined cold treatment. In addition to the transcription factors and genes related to cold tolerance and BR signaling pathway mentioned above, some other genes related to cold tolerance were identified, such as zinc finger domain containing protein, auxin-induced protein, oxygen evolving enhancer protein, heat shock protein, etc. The precise function and regulatory mechanism of these genes still needs further investigation.

In conclusion, many QTLs and candidate genes related to cold tolerance or BR pathway were identified by mapping analysis and qRT-PCR. This study provided a basis for further mining the genes involved in low-temperature tolerance or BR signaling pathway and investigating the mechanism regulating low-temperature tolerance in rice. Further studies were needed to thoroughly investigate whether these genes are associated with both cold tolerance and BR signaling pathway in rice.

## 4. Materials and Methods

### 4.1. Plant Materials and Population Development

A super rice variety SN265 was used as the recipient parent to cross with the donor parent DXWR. F_1_ plants were backcrossed with the recipient parent SN265 for 4 times to construct the BC_4_F_1_ generation. Then, the BC_4_F_1_ population was used to develop a BRIL population with 140 individuals in F_8_ generation (BC_4_F_8_).

### 4.2. Phenotypic Evaluation for Cold Tolerance

Evaluation of cold tolerance at seedling stage was performed according to the previous studies with minor changes [6,29,30]. For breaking dormancy, seeds were placed in a drying oven at 50 °C for 72 h. Then, the surface of the seeds, sterilized in 1% NaClO solution, was germinated and grew in a growth chamber with 75% humidity and 12 h light/12 h dark conditions. When the seedlings grew to the third leaf stage, the seedlings were watered and sprayed with sterilized distilled water (for cold treatment) and 0.1 μmol/L BR solution (for BR-combined cold treatment) for 24 h before the cold treatment, respectively. Furthermore, the seedlings were stressed at 4 °C for 7 days, and then moved to a greenhouse at 25 °C for 7 days to allow the seedlings to resume normal growth. Cold tolerance was evaluated based on the phenotypic changes of seedling shoot length (SSL), seedling root length (SRL), seedling dry weight (SDW), and seedling wet weight (SWW), respectively. SSL, SRL, SDW, and SWW of the abovementioned seedlings after recovering for 7 days were measured. The average of the three replicates of 10 seedlings for each treatment was analyzed. For the cold treatment, the data obtained under normal temperature (25 °C) were used as control. Cold tolerance scores based on the reduction rate of SSL, SRL, SDW, and SWW were calculated as follows: Reduction rate = ((data under normal temperature−data under cold treatment)/data under normal temperature) ×100%. For BR-combined cold treatment, both data obtained under normal temperature and cold treatment were used as controls, respectively. Cold tolerance scores were based on the reduction rate of SSL, SRL, SDW, and SWW = ((data under normal temperature−data under BR-combined cold treatment)/data under normal temperature) × 100% or ((data under BR-combined cold treatment−data under cold treatment)/data under BR-combined cold treatment) × 100%, respectively.

All of the experiments for evaluating cold tolerance were repeated three times under the same conditions, and the average cold tolerance scores from three replicates were used for the QTL mapping analysis. R software 3.5.1 (R Core Team. R: A Language and Environment for Statistical Computing. Vienna: R Foundation for Statistical Computing) was used to analyze the frequency distributions and correlations of all 4 phenotypic traits (www.R-project.org, accessed on 1 July 2022).

### 4.3. Genotyping-by-Sequencing and SNP Identification

Young leaves of 140 BRIL individuals and their parental lines were collected separately for genomic DNA extraction using a DNA extraction kit (Aidlab, Beijing, China). DNA libraries from BRILs were constructed using the GBS technology. The restriction enzymes *Pst*I and *Msp*I and T4 ligase were used for digestion and ligation, respectively. The DNA libraries were enriched and sequenced using Illumina HiSeq 4000 instrument (Huada Gene Technology Co., Ltd., Shenzhen, China). The Tassel software was used to analyze the raw data and high-quality SNPs between parents were termed by alignment with Nipponbare reference genome [53].

### 4.4. Construction of Linkage Map and QTL Analysis

The Illumina data of BRIL population was used to construct a linkage map using the QTL IciMapping sofware v4.163 (Meng et al. 2015, Beijing, China) [54]. The genotypic maps were aligned and split into recombination bins according to the recombination breakpoints, with the parameter of window size of 10 cm and walk speed of 1 cm.

QTLs were mapped using the inclusive composite interval mapping of the additive (ICIM-ADD) mapping method. QTLs were computed by a permutation test involving 1000 runs at a significance level of *p* = 0.05. The threshold for the logarithm of odds (LOD) scores was set to 3.0. QTL nomenclature was followed by the method of McCouch et al. (2008) [55].

### 4.5. Prediction and Validation of Candidate Genes

To predict potential candidate genes within QTL intervals, the physical positions of SNP markers flanking the QTLs were searched in the Rice Genome Annotation Project website (http://rice.plantbiology.msu.edu, accessed on 1 July 2022). The seedlings at the third leaf stage from 2 parental lines, 3 excellent cold tolerance BRIL individuals, and 3 non-excellent cold tolerance BRIL individuals at normal temperature (25 °C), for 12 h under cold stress (4 °C), and for 12 h under BR-combined cold treatment (pre-watering and pre-spraying with 0.1 μmol/L BR solution for 24 h, 4 °C) were collected to extract total RNA using the Trizol reagent. Quantitative real-time PCR (qRT-PCR) was used to investigate the expression patterns of 10 representative candidate genes. Moreover, qRT-PCR was carried out using a q225 Real-Time PCR System (Monad, China) under the following conditions: 95 °C for 5 min, 30 cycles of 95 °C for 15 s, 60 °C for 30 s, and 72 °C for 1 min, and 1 cycle of 72 °C for 15 min. Each sample was analyzed in three biological and three technical replicates. The relative expression levels were calculated via the ∆*C*t method [56]. The actin gene of rice (LOC_Os03g50885) was used as a reference gene. All of the primer sequences are listed in Appendix A.

## Figures and Tables

**Figure 1 plants-11-02324-f001:**
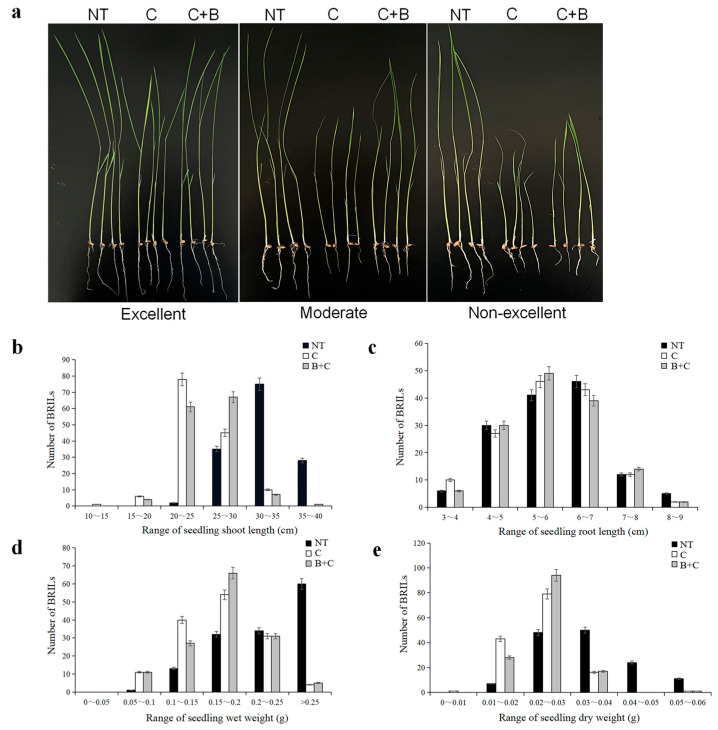
Phenotypic characterization under cold stress and BR-combined cold treatment at seedling stage in BRILs. (**a**) Phenotypes of the representative BRILs with excellent, moderate, and non-excellent cold tolerance; (**b**) ranges of the seedling shoot length (SSL) in BRILs; (**c**) ranges of the seedling root length (SRL) in BRILs; (**d**) ranges of the seedling dry weight (SDW) in BRILs; (**e**) ranges of the seedling wet weight (SWW) in BRILs. NT: Normal temperature; C: Cold stress; C + B: BR-combined cold treatment.

**Figure 2 plants-11-02324-f002:**
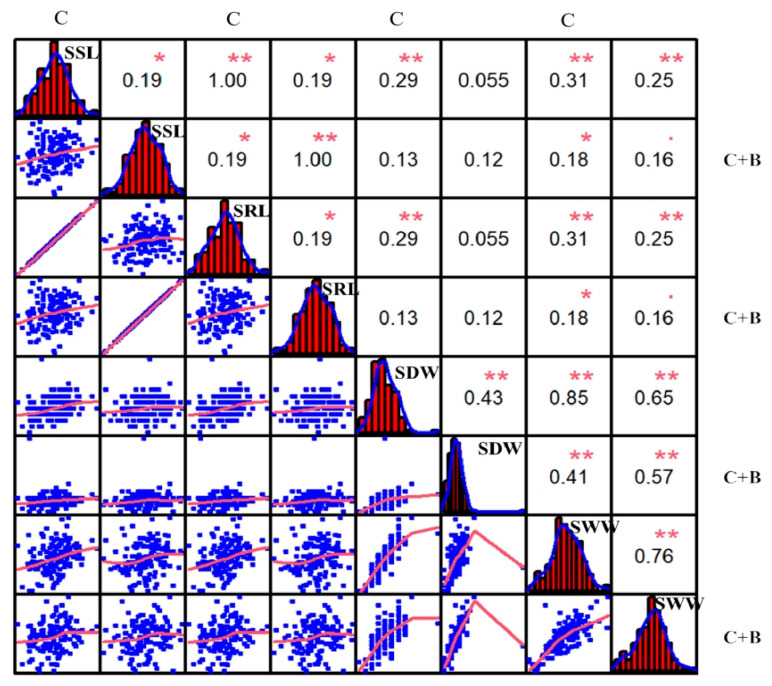
Correlation analysis of SSL, SRL, SDW, and SWW under cold stress and BR-combined cold treatment. Plots on the diagonal line show the phenotypic distribution of each trait as indicated; values above the diagonal line are Pearson’s correlation coefficients between traits; plots below the diagonal line are scatter plots of compared traits. ** *p* ≤ 0.01; * *p* ≤ 0.05; · *p* ≤ 0.1. SSL: Seedling shoot length; SRL: Seedling root length; SDW: Seedling dry weight; and SWW: Seedling wet weight; C: Cold stress; C + B: BR-combined cold treatment.

**Figure 3 plants-11-02324-f003:**
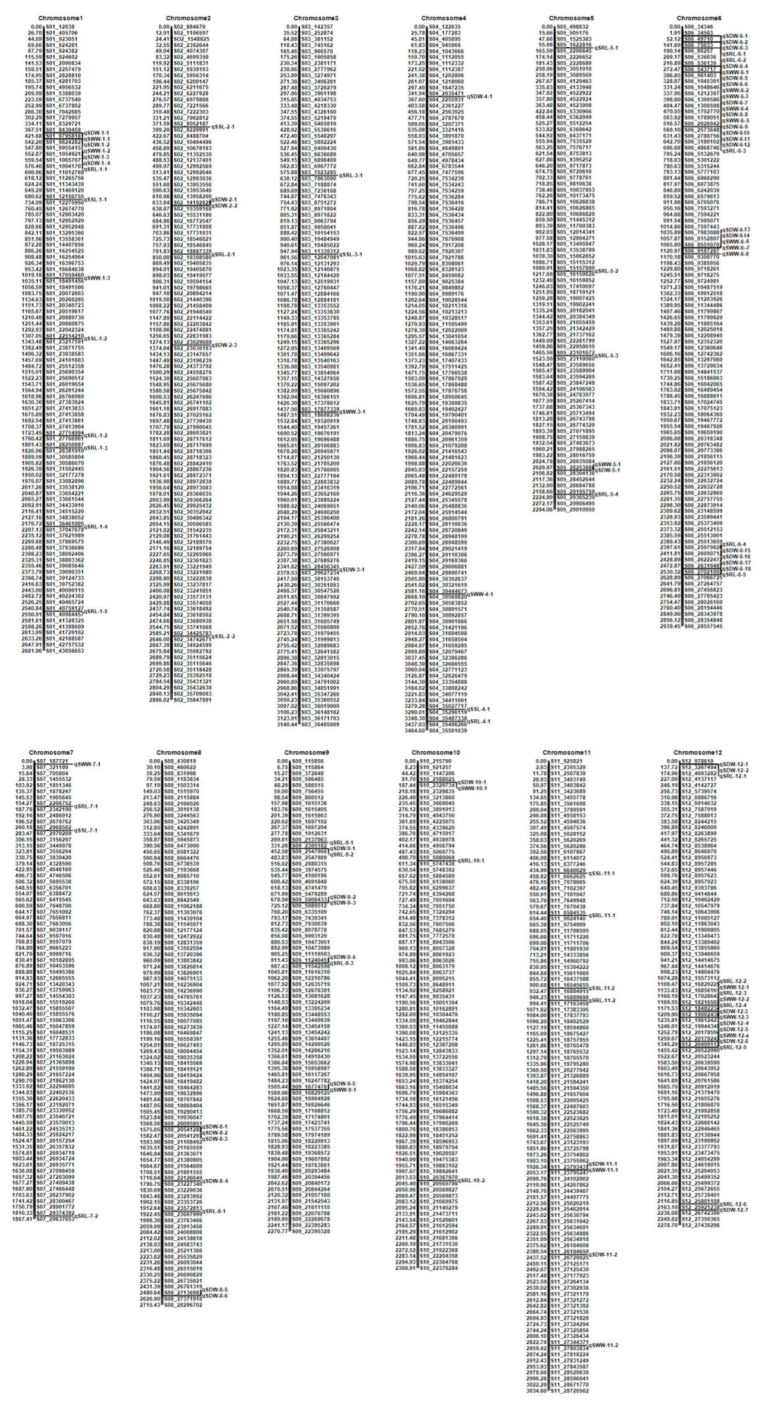
Molecular genetic map showing the positions of QTLs for four traits investigated under cold stress and BR-combined cold treatment.

**Figure 4 plants-11-02324-f004:**
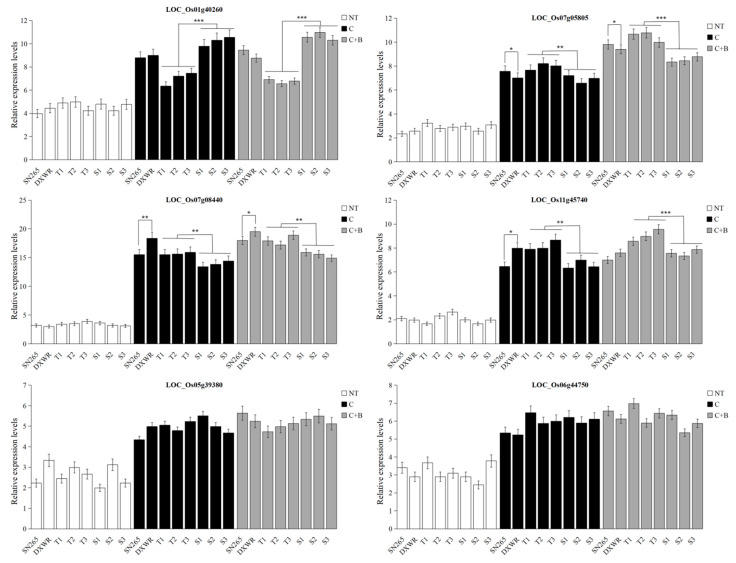
Expression patterns of some candidate genes. Biological triplicates were averaged and statistically analyzed via Student’s *t*-test (* *p* < 0.05; ** *p <* 0.01; *** *p <* 0.001). NT: Normal temperature; C: Cold stress; C + B: BR-combined cold treatment.

**Table 1 plants-11-02324-t001:** Summary of genetic linkage map characteristics in BRILs.

Chr	Number of Bin Markers	Genetic Distance (cM)	Ave. Genetic Distance between Markers (cM)	Ave. Interval (cM)	Interval Range (cM)
1	104	268.20	0.4	25.8	5.1–120.5
2	112	288.60	0.3	25.8	4.0–74.5
3	116	314.04	0.3	27.1	4.9–73.6
4	117	346.46	0.3	29.6	7.5–88.7
5	81	229.41	0.4	28.3	3.6–127.8
6	100	293.94	0.4	29.4	1.9–98.6
7	77	186.74	0.4	24.3	4.0–75.6
8	91	271.54	0.5	29.8	6.3–146.3
9	79	227.08	0.3	28.7	6.2–121.3
10	85	230.99	0.3	27.2	5.4–112.6
11	105	303.46	0.3	28.9	2.0–126.8
12	78	227.87	0.3	29.2	2.0–137.7
All	1145	3188.33	0.3 (Ave.)	27.8 (Ave.)	1.9–146.3

Chr: Chromosome number; Ave.: Average.

**Table 2 plants-11-02324-t002:** QTL summary under different conditions for SSL.

Condition	QTL	Chr	Interval	LOD	PVE (%)	Add
**Cold**	*qSSL1-2*	1	22,514,210–23,217,591	4.7469	9.8918	0.0402
	*qSSL3-1*	3	11,339,312–12,047,081	3.3850	7.7212	−0.0398
	*qSSL11-1*	11	6,640,029–6,662,625	4.8162	8.6890	−0.0356
**BR + C-N**	*qSSL1-1*	1	12,150,755–12,275,956	4.3992	6.3653	−0.0364
	*qSSL2-1*	2	8,052,187–8,229,991	3.6184	10.7271	0.0374
	*qSSL4-1*	4	35,027,717–35,296,119	5.0857	9.4770	0.0401
	*qSSL11-2*	11	16,545,655–16,884,915	3.9168	9.1290	−0.0356
**BR + C-C**	*qSSL2-2*	2	34,425,783–34,742,671	3.9258	15.0490	−0.0537
	*qSSL7-1*	7	23,483,968–23,485,531	3.1103	4.1682	0.0259

Chr: Chromosome number; LOD: Logarithm of odds value; PVE: Phenotypic variance explained; Add: Additive effect; Cold: Cold stress; Cold + BR-N: Data under normal temperature condition were used as the control; Cold + BR-C: Data under cold treatment condition were used as the control.

**Table 3 plants-11-02324-t003:** QTL summary under different conditions for SRL.

Condition	QTL	Chr	Interval	LOD	PVE (%)	Add
Cold	*qSRL10-2*	10	20,367,970–20,569,786	3.0102	7.7200	0.1398
	*qSRL11-2*	11	16,889,698–17,163,458	3.1469	7.6157	−0.1713
BR + C-N	*qSRL1-3*	1	28,250,887–28,381,910	3.0384	9.1064	0.1096
	*qSRL8-1*	8	23,572,813–23,607,999	3.6878	9.4108	0.1096
BR + C-C	*qSRL1-1*	1	9,795,818–9,824,282	4.0158	1.6769	−0.2551
	*qSRL1-2*	1	27,714,994–27,766,901	3.1117	1.1912	−0.1597
	*qSRL1-4*	1	36,461,005–37,047,678	4.0028	0.7550	0.1129
	*qSRL1-5*	1	40,759,127–40,984,457	3.0319	0.9515	0.9515
	*qSRL2-1*	2	18,887,339–19,398,580	3.3044	1.8057	−0.2096
	*qSRL3-1*	3	7,023,285–7,063,090	3.0500	1.6192	−0.2303
	*qSRL4-1*	4	35,487,338–35,496,266	3.2674	1.7908	−0.2234
	*qSRL5-1*	5	1,622,816–2,206,645	3.1717	1.7711	−0.2284
	*qSRL5-2*	5	15,157,600–16,159,826	3.2548	1.7136	−0.2395
	*qSRL5-3*	5	23,101,672–23,110,060	4.1914	1.7021	−0.2340
	*qSRL5-4*	5	29,195,759–29,365,239	3.0047	1.7353	−0.2379
	*qSRL6-1*	6	34,503–49,710	3.2385	1.4471	−0.2590
	*qSRL6-2*	6	49,710–75,033	3.7394	1.5946	−0.2537
	*qSRL6-3*	6	2,026,942–2,573,648	3.3184	1.6972	−0.2214
	*qSRL6-4*	6	26,319,487–27,021,092	4.1196	1.6335	−0.2495
	*qSRL6-5*	6	27,021,092–27,066,725	3.6184	1.6633	−0.2488
	*qSRL7-1*	7	2,206,752–2,342,190	3.1001	1.2549	0.1573
	*qSRL7-2*	7	29,374,392–29,637,653	3.3382	1.5548	−0.2212
	*qSRL9-1*	9	2,137,063–2,395,182	3.3416	1.6710	−0.2369
	*qSRL9-2*	9	2,395,182–2,547,966	3.8906	1.7777	−0.2301
	*qSRL9–3*	9	11,240,451–11,542,598	3.5838	1.6649	−0.2410
	*qSRL10-1*	10	5,080,068–5,747,438	3.0349	1.6997	−0.2413
	*qSRL11-1*	11	8,584,535–9,024,140	3.4737	1.3106	0.2003
	*qSRL12-1*	12	978,610–3,267,494	3.2328	1.6911	−0.2408
	*qSRL12-2*	12	18,216,503–18,482,234	3.1496	1.8460	−0.1963
	*qSRL12-3*	12	18,482,234–19,002,436	4.3086	1.6868	−0.2264
	*qSRL12-4*	12	20,179,243–20,499,113	3.1560	1.6292	−0.2544
	*qSRL12-5*	12	20,499,113–20,522,990	3.0551	1.6346	−0.2470
	*qSRL12-6*	12	25,801,108–25,921,238	4.0588	1.5263	−0.2428

Chr: Chromosome number; LOD: Logarithm of odds value; PVE: phenotypic variance explained; Add: Additive effect; Cold: Cold stress; Cold + BR-N: Data under normal temperature condition were used as the control; Cold + BR-C: Data under cold treatment condition were used as the control.

**Table 4 plants-11-02324-t004:** QTL summary under different conditions for SDW.

Condition	QTL	Chr	Interval	LOD	PVE (%)	Add
**Cold**	*qSDW1-1*	1	8,430,459–9,795,818	3.0094	1.6880	−0.4911
	*qSDW1-3*	1	9,795,818–9,824,282	3.6108	1.7076	−0.4813
	*qSDW2-3*	2	23,029,608–23,036,193	3.2848	1.8389	−0.3107
	*qSDW4-1*	4	2,035,471–2,255,931	3.6631	0.8453	0.1214
	*qSDW5-1*	5	28,253,886–28,304,136	3.2173	1.8218	−0.3146
	*qSDW6-1*	6	34,503–49,710	5.2751	1.8124	−0.3869
	*qSDW6-5*	6	49,710–75,033	4.6802	1.7782	−0.3912
	*qSDW6-8*	6	75,033–86,257	3.9971	1.7254	−0.3676
	*qSDW6-11*	6	543,717–961,403	4.5950	1.7380	−0.3096
	*qSDW6-13*	6	918,728–8,551,070	3.1224	1.7444	−0.3801
	*qSDW6-15*	6	26,319,487–27,021,092	3.5723	1.7885	−0.3040
	*qSDW6-18*	6	27,021,092–27,066,725	3.0443	1.7837	−0.3076
	*qSDW8-4*	8	22,126,649–23,227,340	3.4343	1.9619	−0.2550
	*qSDW8-6*	8	27,136,983–27,371,918	3.5504	1.8118	−0.3283
	*qSDW9-2*	9	5,908,433–5,989,512	3.2579	1.5819	−0.2097
	*qSDW9-4*	9	11,240,451–11,542,598	3.1615	1.8326	−0.2532
	*qSDW12-1*	12	978,610–3,267,494	3.7306	1.8958	−0.3331
	*qSDW12-4*	12	20,179,243–20,499,113	3.0638	1.8157	−0.3170
	*qSDW12-5*	12	20,499,113–20,522,990	3.1036	1.8231	−0.3093
**BR + C-N**	*qSDW1-2*	1	8,430,459–9,795,818	4.1163	1.0370	−0.4150
	*qSDW1-4*	1	9,795,818–9,824,282	9.6158	1.6340	−1.3208
	*qSDW2-2*	2	14,192,829–15,359,168	9.4888	1.6340	1.3208
	*qSDW6-2*	6	34,503–49,710	10.4920	1.6340	−1.3208
	*qSDW6-6*	6	49,710–75,033	11.4737	1.6340	−1.3208
	*qSDW6-7*	6	75,033–86,257	9.8876	1.6340	−1.3208
	*qSDW6-9*	6	536,139–543,717	9.1114	1.6340	−1.3208
	*qSDW6-12*	6	543,717–961,403	9.1993	1.6340	−1.3208
	*qSDW6-14*	6	8,551,070–9,187,287	10.4330	1.6340	−1.3208
	*qSDW6-16*	6	26,319,487–27,021,092	3.3065	0.9447	−0.3687
	*qSDW6-17*	6	27,021,092–27,066,725	8.0183	1.6340	−1.3208
	*qSDW8-1*	8	20,095,951–20,541,282	4.6413	1.6340	1.3208
	*qSDW8-5*	8	27,136,983–27,371,918	8.6442	1.6340	−1.3208
	*qSDW9-1*	9	2,395,182–2,547,966	7.9756	1.6340	−1.3208
	*qSDW9-3*	9	5,908,433–5,989,512	3.0530	0.4007	−0.1190
	*qSDW9-5*	9	16,774,761–16,829,125	8.6080	1.6340	−1.3208
	*qSDW10-1*	10	2,188,025–2,329,734	8.8075	1.6340	−1.3208
	*qSDW11-2*	11	26,184,650–26,729,025	8.8993	1.6340	−1.3208
	*qSDW12-2*	12	978,610–3,267,494	8.3168	1.6340	−1.3208
	*qSDW12-3*	12	20,179,243–20,499,113	8.3537	1.6340	−1.3208
	*qSDW12-6*	12	20,499,113–20,522,990	8.0866	1.6340	−1.3208
**BR + C-C**	*qSDW2-1*	2	14,192,829–15,359,168	9.2210	6.5896	0.9641
	*qSDW3-1*	3	28,456,345–29,627,234	3.9577	1.7956	−0.1035
	*qSDW6-3*	6	34,503–49,710	6.6853	4.5867	−0.8043
	*qSDW6-4*	6	49,710–75,033	6.5255	4.5867	−0.8043
	*qSDW6-10*	6	536,139–543,717	5.2886	4.5867	−0.8043
	*qSDW8-2*	8	20,095,951–20,541,282	3.9832	6.0808	0.9261
	*qSDW8-3*	8	20,095,951–20,541,282	3.6278	1.6461	0.1350
	*qSDW11-1*	11	23,793,431–23,795,247	4.9421	4.5867	−0.8043
	*qSDW12-7*	12	25,921,238–26,742,306	4.0396	1.8563	0.1470

Chr: Chromosome number; LOD: Logarithm of odds value; PVE: Phenotypic variance explained; Add: Additive effect; Cold: Cold stress; Cold + BR-N: Data under normal temperature condition were used as the control; Cold + BR-C: Data under cold treatment condition were used as the control.

**Table 5 plants-11-02324-t005:** QTL summary under different conditions for SWW.

Condition	QTL	Chr	Interval	LOD	PVE (%)	Add
Cold	*qSWW1-3*	1	17,059,460–18,491,456	8.4626	18.2439	0.1201
	*qSWW7-1*	7	187,721–321,199	4.1511	8.7113	−0.1427
	*qSWW12-2*	12	19,002,436–19,612,427	3.9255	7.5760	0.0745
BR + C-N	*qSWW3-1*	3	17,877,339–18,608,236	3.7539	2.3749	0.1061
	*qSWW6-2*	6	49,710–75,033	5.2304	5.4240	−0.7790
	*qSWW6-3*	6	75,033–86,257	3.6894	5.4240	−0.7790
	*qSWW6-6*	6	8,551,070–9,187,287	4.6827	5.4241	−0.7790
	*qSWW11-2*	11	27,344,371–27,803,834	4.9243	3.2351	0.1113
BR + C-C	*qSWW1-1*	1	8,430,459–9,795,818	3.9405	0.7179	−0.6502
	*qSWW1-2*	1	9,795,818–9,824,282	4.2951	0.7179	−0.6501
	*qSWW4-1*	4	30,444,673–30,569,820	3.1941	0.7601	−0.4591
	*qSWW5-1*	5	28,253,886–28,304,136	3.1392	0.8057	−0.3553
	*qSWW6-1*	6	49,710–75,033	5.6121	0.7489	−0.4876
	*qSWW6-4*	6	75,033–86,257	4.0525	0.7134	−0.2935
	*qSWW6-5*	6	536,139–543,717	3.3069	0.7941	−0.3672
	*qSWW6-7*	6	8,551,070–9,187,287	5.4922	0.7179	−0.6503
	*qSWW6-8*	6	9,187,287–9,300,770	3.5735	0.7532	−0.4835
	*qSWW9-1*	9	16,774,761–16,829,125	4.1466	0.8111	−0.3491
	*qSWW10-1*	10	2,188,025–2,329,734	3.4112	0.7576	−0.4731
	*qSWW11-1*	11	23,793,431–23,795,247	3.4724	0.7573	−0.4714
	*qSWW12-1*	12	18,482,234–19,002,436	4.0664	0.7574	−0.4714
	*qSWW12-3*	12	20,179,243–20,499,113	3.5313	0.7576	−0.4694
	*qSWW12-4*	12	20,499,113–20,522,990	3.3591	0.7582	−0.4672

Chr: Chromosome number; LOD: Logarithm of odds value; PVE: Phenotypic variance explained; Add: Additive effect; Cold: Cold stress; Cold + BR-N: Data under normal temperature condition were used as the control; Cold + BR-C: Data under cold treatment condition were used as the control.

**Table 6 plants-11-02324-t006:** Summary of the annotated candidate genes possibly associated with cold tolerance or BR signaling pathway.

Condition	QTL	Chr	Locus Name	Gene Coordinates	Gene Product
C	*qSSL1-2*	1	LOC_Os01g40260	22,731,943–22,733,237	OsWRKY77- Superfamily of TFs with WRKY and zinc finger domains
C + BR	*qSWW11-2*	11	LOC_Os11g45740	27,670,321–27,673,334	MYB family transcription factor
C + BR-N	*qSSL7-1*	7	LOC_Os07g05805	23,483,968–23,485,531	OsBZR1, transcription factor, Brassinosteroid (BR)-regulated growth response
C	*qSSL7-3*	7	LOC_Os07g08440	4,338,514–4,342,219	bHLH transcription factor
C/C + BR-N	*qSDW6-18/17*	6	LOC_Os06g44750	27,025,437–27,029,339	AP2 domain-containing protein, expressed
C + BR-N/C + BR-C	*qSDW6-9/10*	6	LOC_Os06g01966	543,057–545,780	auxin-induced protein 5NG4, putative, expressed
C + BR-C	*qSRL12-6*	12	LOC_Os12g41820	25,901,456–25,907,573	heat shock protein DnaJ, putative, expressed
C	*qSSL7-1*	7	LOC_Os07g01480	306,009–307,555	oxygen evolving enhancer protein 3 domain-containing protein, expressed
C + BR-C	*qSRL12-6*	12	LOC_Os12g41700	25,815,277–25,818,874	LSD1, zinc finger domain-containing protein, expressed
C + BR-C	*qSRL5-3*	5	LOC_Os05g39380	23,101,671–23,104,153	zinc finger, C3HC4 type domain-containing protein, expressed
C + BR-C	*qSWW9-1*	9	LOC_Os09g27650	16,822,234–16,825,686	ZOS9-14, C2H2 zinc finger protein, expressed
C + BR-N	*qSDW9-5*	9	LOC_Os09g27660	16,829,409–16,837,307	OsFBO21, F-box, and other domain-containing protein, expressed
C + BR-N	*qSDW10-1*	10	LOC_Os10g04590	2,185,226–2,188,547	OsFBX358, F-box domain-containing protein, expressed

Chr: Chromosome number; C: Cold treatment; C + BR-N: Data under normal temperature condition were used as the control; C + BR-C: Data under cold treatment condition were used as the control.

## Data Availability

The raw data supporting the conclusions of this manuscript can be found in the National Center for Biotechnology Information (NCBI) BioProject, https://www.ncbi.nlm.nih.gov/bioproject (accessed on 10 May 2022), PRJNA836720.

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
