# Peer review of "Identification of QTL under Brassinosteroid-Combined Cold Treatment at Seedling Stage in Rice Using Genotyping-by-Sequencing (GBS)"

_plants, 2022, doi:10.3390/plants11172324_

Round 1

Reviewer 1 Report

In this work, QTL and cold tolerance genes alone and in combination with BR response were identified. For this, a group of inbred lines derived from a 4-generation backcross with an ancestor of the cultivated rice was used. This work provides important useful information for breeders, but its presentation, such as the extension of the data, is not adequate, making it very difficult to read and, furthermore, a more stringent analysis could be made in order to adjust the important data, before to be considered for publication in this journal.

If the cold tolerant donor parent is DXWR, but, the work is realized with 140 inbred line that procceded of four back crossing with the cold sensitive donor parent SN265 and any selection by cold tolerance was done in the process, Why would this material be suitable for this type of experiments?. Relevant tolerance genes may have been lost.

QTL score LOD setted was 3, why this LOD value was choosen?. The recent bibliography in GWAS in rice used LOD value 4 (p 0.0001). This value would decrease the number of QTL and would allow obtaining more significant results, in addition to simplifying the writing and presentation of data.

tauthors should analyze their results with greater astringency.

What was the selection criteria for the 10 genes evaluated by real-time PCR?.

Figure 3 is unreadable.

Reviewer 2 Report

The manuscript is well-written, marginal comments are marked in the text.

Author Response

Point 1: The reviewer 2 pointed out some spelling errors and format errors such as use of italic.

 Response 1: We thank the reviewer’s contributions to the peer review process of our manuscript and we are glad that the reviewer offered affirmation to our work. According to the reviewer’s helpful suggestion, all spelling and format errors have been modified.

Round 2

Reviewer 1 Report

the authors have adequately answered the questions posed